# Online learning success model for adults in open and distance education in Western China

**Li Yuebo**[1], **Siti Hajar Halili**[2]*, **Rafiza Abdul Razak**[2]

1 Enrollment and Employment Guidance Center, Guizhou Open University, Guiyang, Guizhou, China,
2 Department of Curriculum and Instructional Technology, Faculty of Education, Universiti Malaya, Kuala Lumpur, Malaysia

* siti_hajar@um.edu.my

**Data Availability Statement:** All relevant data are within the manuscript and its Supporting Information files.

**Funding:** This work was specifically funded by Universiti Malaya under the grant number

## Abstract

This study investigates the factors influencing online learning (OL) success among non-full-time adult students in open and distance education in Western China. To utilize a structural equation model with seven construction elements that combine the information system success (ISS) model and TPACK theory. Data from 245 participants were analyzed using PLS-SEM. Results reveal that system quality, service quality, and teachers' TPACK ability have varying degrees of positive impact on OL success. The main contribution of this study is its innovative combination of TPACK theory and the ISS Model, which has not been extensively explored in previous research. Additionally, this study emphasizes the significance of addressing the distinctive requisites and attributes of part-time adult learners engaged in online learning (OL). The findings of this study can help educational practitioners and policymakers create more effective and efficient OL environments that meet the needs of adult learners and bridge the gap between theory and practice.

## Introduction

The rapid development of information and communication technology (ICT) has led to the emergence of online education as a flexible and accessible alternative to traditional classroom-based learning. Online learning has gained significant traction in open and distance education (ODE) institutions in China, providing non-full-time adult students with opportunities to pursue education while managing other commitments. However, despite its potential benefits, there are challenges and barriers to the success of online learning, particularly for non-full-time adult students [1–3].

China's Open and Distance Education (ODE) system, spearheaded by the National Open University, has significantly expanded educational access, empowering the public to augment their knowledge and academic credentials. The onset of the COVID-19 pandemic further expedited the uptake of online learning, prompting the complete shift of adult education from offline to online delivery at the Open University of China (OUC). Even post-pandemic, this mode of education persisted. The allure of online learning for adult students lies in its

UMG0030-2021 (UM.0000412/HGA.GV) and was awarded to Dr. Siti Hajar Halili.

**Competing interests:** The authors have declared that no competing interests exist.

adaptable nature concerning time and space, affording them the autonomy to select learning materials and program structures tailored to their distinct needs and circumstances. This flexibility is particularly advantageous for part-time students who juggle professional responsibilities alongside their studies. [4–6].

However, despite the benefits, non-full-time adult learners face various challenges that impact their learning experience. Balancing work, family, and other life commitments can impede their ability to fully engage in learning activities. Insufficient institutional support and instructional methods that do not cater to the specific circumstances of non-full-time adult students can lead to poor learning outcomes [7].

To address these challenges and enhance online learning success, it is essential to identify the factors that influence the effectiveness of online learning. The information system success (ISS) model has been widely used to study the success of online learning, examining factors such as technology system quality, information quality, service quality, support system quality, learner quality, teacher quality, and perceived usefulness [8–10]. However, most of these studies have focused on universities in developed countries.

In recent years, researchers have made significant advancements in the field of online learning by expanding the existing Information System Success (ISS) model and integrating it with other relevant theories. These include the Technology Acceptance Model (TAM) and the Expectation Confirmation Model (ECM) [11–23]. These studies have explored various factors that contribute to user-perceived satisfaction, usage, and benefits of e-learning.

Furthermore, researchers have also investigated the influence of non-cognitive abilities and cultural factors on the success of online learning [13, 24]. Notably, [11] proposed a comprehensive online learning success model that encompasses multiple dimensions, providing a more thorough evaluation of the effectiveness of online learning systems. This model specifically examined the technological competence of teachers and the impact of their attitudes toward technology use on student learning outcomes.

Despite these advancements, limited research has explored the potential of the Technological Pedagogical and Content Knowledge (TPACK) framework for teachers, which emphasizes the intersection of technology, pedagogy, and content knowledge. This framework offers a comprehensive approach to understanding and enhancing teachers' abilities and qualities in the context of online learning. Therefore, further exploration of the TPACK framework is crucial in order to fully understand the complex dynamics of online learning success and its implications for instructional design and implementation.

This study aims to bridge these research gaps by developing an online learning success model for non-full-time adult students in open and distance education in Western China. By integrating the TPACK framework with the ISS model, this study sought to identify the factors that positively influenced online learning success and provided insights for improving the quality of distance and open education in China. Specifically, this research focused on analyzing non-full-time adult students in Western China as research subjects to verify the proposed online learning success model. By collecting and analyzing data from 245 participants using Partial Least Squares Structural Equation Modeling (PLS-SEM), this study aimed to examine the impact of system quality, service quality, and teachers' TPACK ability on online learning success.

The main contribution of this study lies in its innovative combination of the TPACK framework and the ISS model, which has not been extensively explored in previous research. By incorporating teachers' technological, pedagogical, and content knowledge into the model, this study provides a more comprehensive understanding of the factors that influence online learning success. Additionally, this research emphasizes the importance of considering the specific needs and characteristics of non-full-time adult students in online learning.

The findings of this study have practical implications for educational practitioners and policymakers in designing and implementing effective and efficient online learning environments. By addressing the identified factors, institutions can better cater to the needs of adult learners, bridge the gap between theory and practice, and enhance the overall success of online learning initiatives.

This study investigated the factors influencing online learning success among non-full-time adult students in open and distance education in Western China. By integrating the TPACK framework with the ISS model, this research contributes to the existing literature by providing a more comprehensive understanding of the determinants of online learning success. The findings of this study can inform the development of strategies and interventions to create more effective and efficient online learning environments that meet the specific needs of adult learners.

## Conceptual model of this study

This study delves into the net benefits of online learning, focusing on individuals' evaluations of "perceived satisfaction with the use of an online learning system" and their "intention to continue using an online learning system" after engaging with online learning for academic or career progression. The research is grounded in an assessment of theoretical models of adult online learning instruction employed across various programs within the Chinese educational context. This conceptual framework integrates the ISS Model by [25] along with [26] TPACK theory. The conceptual model is shown in Fig 1. A literature review found that the theories evaluated in this research were valid, reliable and have a solid theoretical and pedagogical basis.

## Constructs identified in this study

This research model encompasses nine constructs: system quality (SQ), service quality (SEQ), information quality (IQ), continuous use (USE), user-perceived satisfaction (US), TPACK,

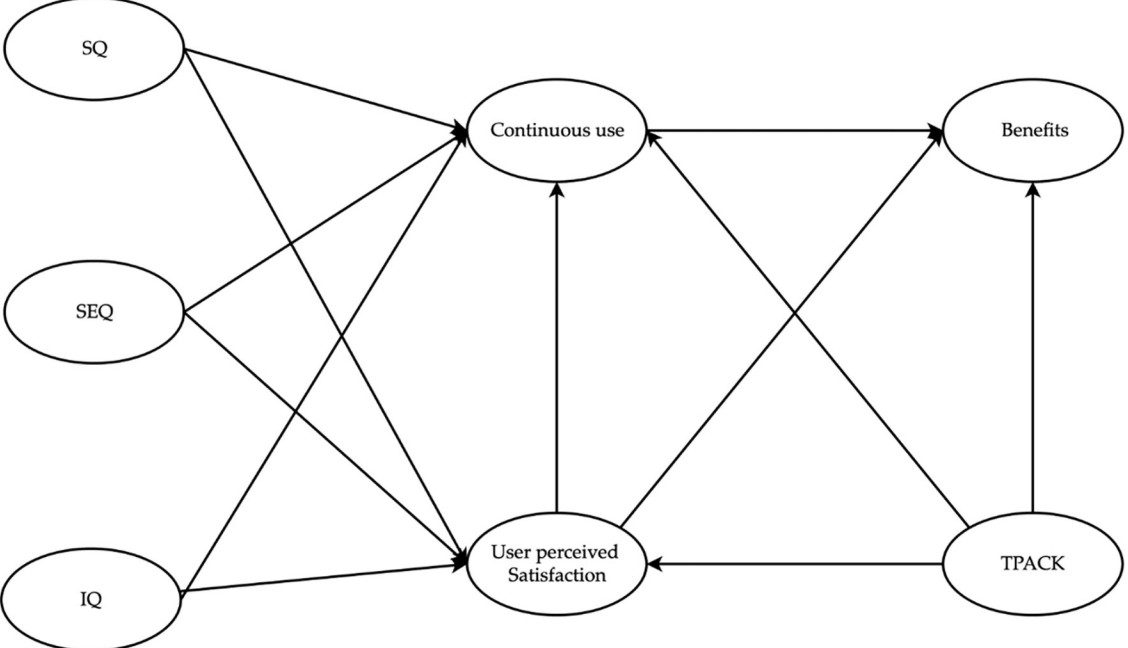

**Fig 1. Proposal for a model of success in adult online learning.**

and net benefit (NB). The effectiveness of an online learning (OL) system pivots on the level of support and efficiency it extends to users during their platform engagement. Elevated system quality not only amplifies user convenience on the network platform but also fortifies user privacy and expedites online information retrieval [11]. Service quality, as delineated by [27], encapsulates user expectations and requirements when interacting with an OL platform. Information quality, outlined by [25], encompasses the integrity, timeliness, accuracy, relevance, and stability of provided information.

Moreover, learners' willingness to consistently engage with an OL platform aligns with the concept of USE [25], while their cognitive appraisal to assess the reasonableness of efforts and benefits derived from the system is denoted as US [25]. NB pertains to learners' academic, career, and personal development post-learning [28].

Teachers are expected to acquire TPACK, which represents the essential knowledge required for effectively amalgamating teaching content, instructional techniques, and information technology [26]. It is more practically the pedagogical approach of the instructor when technology is utilized in the delivery of instruction.

The responsibility for implementing TPACK lies with educators, necessitating a deliberate focus on the active involvement of classroom teachers in the design and execution of effective pedagogical strategies within technologically enriched teaching environments. Both teachers and administrators assume pivotal roles in overseeing the critical self-assessment of classroom instruction. Within teacher education programs, TPACK training should underscore the fusion of technology in its diverse forms as an integral component of a dynamic approach to crafting and executing pedagogical methods tailored for adult learners.

TPACK encompasses three fundamental knowledge domains: subject content, instructional methodology, and technology. It transcends mere aggregation of these elements, demanding a nuanced comprehension of transmitting knowledge within a high-tech educational milieu. A teacher's task involves not only integrating technology into pedagogical approaches but also cultivating a profound understanding of knowledge dissemination from educator to adult learner. The study and application of TPACK must transcend a unilateral emphasis on technology, incorporating teaching and learning theories alongside pedagogical strategies in the design and implementation of internet-based programs.

### Research hypotheses

Online learning (OL) relies on digital systems, and the functionality and user experience of these systems can significantly impact students' attitudes and motivation to actively engage in online learning [29]. According to [30], the implementation of suitable learning systems can enhance learning outcomes. They also highlight the benefits of multimedia and information technology in enriching the online learning experience. Additionally, [31, 32] suggest that utilizing smart devices and virtual communities can facilitate student-led learning. As a result, the following hypotheses have been suggested for this investigation.

H1a: System quality positively influences OL system's continued usage intention.

H1b: System quality significantly impacts perceived satisfaction.

[33] explored from the perspective of customer perceived value and identified empathy, reliability, product mix, ease of use, and security as five key dimensions to measure the quality of online services. The critical factor in whether an online platform can attract and retain users is whether the services provided meet users' needs without face-to-face interaction. This paper argues that good service quality is the ability to provide personalized services to users, as well as the ability of the platform to provide timely solutions for users who encounter difficulties in

using it, all of which influence users' evaluation of whether the online platform is useful and easy to use [34]. As a result, the following hypotheses have been suggested for this investigation.

H2a: Service quality has a significant positive effect on continuously using the OL system.

H2b: Service quality has a significant positive effect on perceived satisfaction.

[25] utilized five dimensions—accuracy, relevance, completeness, timeliness, and consistency—to gauge information quality. Their choice of dimensions has been widely referenced by numerous scholars investigating website information quality. Previous research has shown that information quality is an important criterion for the quality of online platforms, and [35] found that all three dimensions of website quality had a significant impact on the usefulness and ease of use of mobile devices in his study of website quality in the mobile environment. [29] argue that when users use online learning services, their perceived usefulness of course content increases, and their expectation confirmation level increases, if they find the content in the platform they are using to be rich and of high quality. In addition, their level of satisfaction with the course content increases. [36] conducted research on the continuous usage behavior of MOOC users. As part of his investigation, he measured the content quality of MOOC platforms in terms of lecture quality and supplementary learning. He discovered that content quality had a significant positive effect on perceived usefulness and expected confirmation level. Drawing from the findings of both studies, the term "Quality of Content" (QoC) will be used in the context of this study to refer to the quality of the course content on the online education platform. This includes the richness of the course content, the quality of the recording of the course videos, the organization of the course content, and the richness of the course-related resources. The amount of course material quality that users of online education platforms perceive has an effect on both their perception of the utility of the platform they use and the level of confirmation they anticipate receiving as a result of their usage of the platform.

H3a: Information Quality positively impacts perceived satisfaction.

H3b: Information Quality positively affects continuous use.

It was found that teachers' pedagogical and technical skills were enhanced, which to a certain extent, contributed to the effectiveness of their online teaching. The higher the level of TPACK of online teachers, the better the results, which is closely related to their intrinsic motivation. In addition, technology is an important tool for online teaching and learning, and teachers' motivation to use technology is an intrinsic motivator for integrating technology with teaching and learning to achieve more effective teaching and learning [36–39]. In light of this research, the following hypotheses have been proposed.

H4a: TPACK positively influences perceived satisfaction.

H4b: TPACK positively impacts continuous use of the OL system.

H4c: TPACK positively affects net benefits.

The most widely used scale is constructed by contextualizing and adapting the assessment of user satisfaction, which is most variable based on findings from prior research. [40] and subsequent research has proven that contentment is the primary element that influences users' intentions to continue using. This was found in the expectation confirmation model (ECM) [41, 42]. As a result, the following hypothesis is put forth: A user's intention to continue using the online education platform they currently utilize will be stronger if the user has a higher level of satisfaction with their evaluation of their experience and their experience using the platform. The following hypothesis is thus formulated.

H5: User perceived satisfaction positively effect on continuous using OL system.

The attitude of online participation is the satisfaction of online participation. In contrast, the depth and breadth of online participation are the behavioral performance, and the continuous use of students is the behavioral performance of learning. The validity of online participation is cognitive performance, or cognitive outcome performance is the learning effect. Obviously, without the satisfaction attitude of participation, there is no sustainable participation behavior or participation in the effectiveness of sustainable behavior. Different from traditional classroom education, online education is a learner-centered autonomous learning process with the media of the Internet and other information technologies. Therefore, the benefits of OL is mainly influenced by OL satisfaction. Continuous use also positively impacts benefits [11, 43–45]. The following hypothesis is thus formulated.

H6: User-perceived satisfaction significantly enhances net benefits.

H7: Continuous use positively contributes to net benefits.

## Methodology

In this study, Partial Least Squares Structural Equation Modeling (PLS-SEM) was employed as the statistical analysis approach. PLS-SEM was chosen for several reasons. First, it allows for the simultaneous assessment of the measurement model and the structural model, which is crucial for examining the relationships between latent variables and their impact on online learning success. Second, PLS-SEM is well-suited for exploratory research and accommodates smaller sample sizes, making it appropriate for the study's sample of 245 participants. Additionally, PLS-SEM can handle both reflective and formative measurement models, capturing the multidimensional nature of the constructs in the research model [46]. Furthermore, PLS-SEM provides robustness against issues such as non-normality and the presence of outliers, which are commonly encountered in survey-based research [47, 48]. Overall, the use of PLS-SEM allowed for the effective analysis of the relationships between system quality, service quality, teachers' TPACK ability, and online learning success in the context of non-full-time adult students in open and distance education in Western China.

[49] delineated the data analysis section's components, comprising both a measurement model and a structural model. The measurement model evaluates the reliability and validity of the variables utilized in this research. Reliability, gauged through Cronbach's alpha and CR values, underwent assessment to ascertain the consistency of the variables. Concurrently, validity examination focused on both discriminative and convergent aspects of the adopted variables. Discriminant validity was assessed using three methods—Fornell-Larcker, HTMT, and Cross loading—to ensure the distinctiveness of the variables. Convergent validity primarily relied on the Average Variance Extracted (AVE) value.

In the structural model, hypothesis testing relied on the bootstrap method to validate the proposed hypotheses. The study encompassed a random sample of 245 participants drawn from five study centers affiliated with a Municipal branch campus of the Open University in western China. The survey, conducted in December 2022, incorporated online questionnaires and on-site paper questionnaires. A total of 245 responses met the criteria for validity, with participants providing written informed consent prior to participation in the study.

### Instrument

Over the past decade, an extensive review and categorization of literature pertaining to attitudes toward online learning (OL), factors contributing to OL quality, and Technological

Pedagogical Content Knowledge (TPACK) was conducted to compile relevant studies. The initial scale construction began by collecting existing research scales. Subsequently, expert interviews were utilized to further refine and augment this scale.

To ensure the accuracy of the Chinese scales in comparison to their English counterparts and to enhance precision, a rigorous two-way translation process was adopted in this study. Furthermore, a group of twenty randomly selected school learners participated in reviewing and completing the questionnaire, identifying any inconsistencies or misconceptions. Based on their feedback, modifications and enhancements were made to the original 5-point Likert scale, resulting in the final scale tailored for this study. items with their sources are followed in S1 Appendix.

**Data collection and analysis.** In December 2022, a questionnaire survey was administered across five study centers affiliated with a municipal branch campus of the Open University in western China. The survey method employed a blend of online and on-site paper questionnaires, yielding an initial response count of 263. Following meticulous scrutiny to ensure response validity, a total of 245 responses were deemed valid and subsequently utilized in the analysis. The socio-demographic profile of the respondents, encompassing gender, age, ongoing education, major, current study semester, occupation, and income, is presented in Table 1. This data offers valuable insights into the sample characteristics.

Regarding the chosen sampling technique, random sampling was employed to ensure the representativeness of the sample and to minimize selection bias. Random sampling allows for the equal probability of selection for each potential participant, increasing the likelihood of obtaining a diverse and unbiased representation of the target population. By employing this technique, the generalizability of the findings was enhanced.

Moreover, the sample size of 245 participants was considered appropriate for this study. This sample size provided a sufficient number of observations for robust statistical analysis and meaningful conclusions. It also offered a reasonable representation of the target population, considering the nature of this research. The chosen sampling technique and sample size were justified in this study as they contributed to the reliability and validity of the findings. These methodological choices aimed to enhance the generalizability of the study's findings to the wider population of part-time adult learners engaged in open and distance education in Western China.

In terms of data analysis, Structural Equation Modeling (SEM) was employed as a multivariate statistical technique. SEM is a multivariate statistical technique that investigates and analyzes complicated data based on the variables' covariance matrices. This method considers measurement errors in both independent and dependent variables, integrating them into the path diagram. This approach facilitates a more precise examination of relationships among observable variables that may not be directly measurable. Among the methodologies within latent variable analysis, the Partial Least Squares Structural Equation Modeling (PLS-SEM), employing partial least squares, is utilized to examine the interactions between independent and dependent variables. Particularly, in scenarios where strong relationships exist between the components, PLS-SEM emerges as an effective analytical approach [46]. The data analysis technique employed in this study involves the use of PLS-SEM3.0 [47, 50] to assess the proposed conceptual framework for success in adult online learning.

## Results

### Common method variance (CMV)

An exploratory factor analysis of the latent variables in this study indicated that the variance explained by the first factor prior to un-rotation was 12.87%, falling below 50% as suggested by

**Table 1. Socio-demographic profile of respondents.**

| | | Frequency | Valid Percent |
|---|---|---|---|
| Gender | 1 | 87 | 35.5% |
| | 2 | 158 | 64.5% |
| Total | | 245 | 100.0% |
| Age | 1 | 4 | 1.6% |
| | 2 | 66 | 26.9% |
| | 3 | 125 | 51.0% |
| | 4 | 41 | 16.7% |
| | 5 | 9 | 3.7% |
| Total | | 245 | 100.0% |
| Education in progress | 1 | 121 | 49.4% |
| | 2 | 124 | 50.6% |
| Total | | 245 | 100.0% |
| Major | 1 | 120 | 49.0% |
| | 2 | 15 | 6.1% |
| | 3 | 12 | 4.9% |
| | 4 | 11 | 4.5% |
| | 5 | 25 | 10.2% |
| | 6 | 40 | 16.3% |
| | 7 | 22 | 9.0% |
| Total | | 245 | 100.0% |
| Which semester are you studying now | 1 | 13 | 5.3% |
| | 2 | 57 | 23.3% |
| | 3 | 70 | 28.6% |
| | 4 | 88 | 35.9% |
| | 5 | 9 | 3.7% |
| | 6 | 8 | 3.3% |
| Total | | 245 | 100.0% |
| Occupation | 1 | 56 | 22.9% |
| | 2 | 21 | 8.6% |
| | 3 | 64 | 26.1% |
| | 4 | 25 | 10.2% |
| | 5 | 14 | 5.7% |
| | 6 | 20 | 8.2% |
| | 7 | 4 | 1.6% |
| | 8 | 2 | 0.8% |
| | 9 | 39 | 15.9% |
| Total | | 245 | 100.0% |
| Income (RMB) | 1 | 38 | 15.5% |
| | 2 | 15 | 6.1% |
| | 3 | 9 | 3.7% |
| | 4 | 21 | 8.6% |
| | 5 | 72 | 29.4% |
| | 6 | 90 | 36.7% |
| Total | | 245 | 100.0% |
| Total | | 245 | 100.0% |

previous research [51, 52]. This finding suggests that there isn't a significant common method bias issue in this paper, minimizing potential impact on subsequent empirical analyses.

## Assessment of the measurement model

Factor loadings of IQ, SQ, SEQ, US, and USE, as both first-order and second-order factors within the constructs of net benefits (NB) and TPACK, demonstrated values surpassing 0.7 for Cronbach's α, rho_A, and CR, indicating strong reliability within these constructs. Furthermore, their Variance Inflation Factor (VIF) values were below 5, indicating negligible multicollinearity.

Each construct exhibited loadings higher than 0.5 for the same item, ensuring strong construct validity by having only one factor with a loading greater than 0.5 for each item within its respective construct. To enhance the scale's validation, standardized loadings for each variable and the Average Variance Extracted (AVE) were examined. Every measured item in this study demonstrated factor loadings exceeding 0.750, all of which held statistical significance (as factor loadings should not be lower than 0.5). Additionally, the AVE for each latent variable surpassed 0.5, affirming the scale's robust convergent validity. The results as shown in Table 2.

The square root of the AVE is compared with the matrix of correlation coefficients between two potential variables to determine which is more meaningful. In assessing discriminant validity between latent variables, a criterion established by [48] suggests that when the squared value of the Average Variance Extracted (AVE) for a latent variable exceeds the correlation coefficients between that variable and others, it signifies sound discriminant validity. In this study, each diagonal value of the AVE exceeded the correlation coefficients between respective latent variables, indicating robust discriminant validity. For instance, the AVE squared for IQ was 0.852, surpassing its correlations with other variables (0.088, 0.467) as depicted in Table 3, affirming the adequacy of these variables' validity. Subsequently, the HTMT method and cross-loading were employed to further assess discriminant validity among the variables. The results indicate a high discriminant validity between them, as shown in Tables 4 and 5.

The structural model assessment The structural model assessment involved parameter estimation using the Bootstrapping method, generating 5000 analogous samples for the analysis. T-values for the path coefficients were calculated in accordance with recommendations by scholars [49]. As depicted in Table 6, the findings indicate that hypotheses H2a, H3a, and H3b are not supported, while the study substantiates the remaining nine hypotheses. As presented in Table 6, hypotheses H2a, H3a, and H3b were not supported, while nine other hypotheses gained support from the findings.

## Discussion

The outcomes substantiate hypotheses H1a and H1b, demonstrating that online learning (OL) system quality positively impacts learners' perceived satisfaction and continuous use. These results align with earlier studies by [9, 11, 12, 14, 53]. It suggests that the OL system's structure, encompassing its reasonableness, ease of use, and flexibility, significantly influences student satisfaction and their willingness to persist in using the system.

Furthermore, the findings validate hypothesis H2b, demonstrating a positive influence of service quality on perceived satisfaction. This result is in line with previous research by [9, 21, 54–56]. It underscores the pivotal role of service quality in shaping students' perceived satisfaction. As online learning support services gain prominence in the competitive education market, they play a crucial role in enhancing educational brands and competitiveness.

However, contrary to expectations and hypothesis H2a, the effect of service quality on continuous use was found insignificant. This finding is consistent with prior studies by [15, 22,

**Table 2. Reliability and convergent validity.**

| Construct | item | loading | VIF | Cronbach's alpha | CR | AVE |
|---|---|---|---|---|---|---|
| NB | AD | 0.880 | 2.155 | 0.875 | 0.923 | 0.800 |
| | CD | 0.910 | 2.685 | | | |
| | PD | 0.893 | 2.403 | | | |
| IQ | IQ1 | 0.879 | 2.843 | 0.873 | 0.914 | 0.726 |
| | IQ2 | 0.875 | 2.981 | | | |
| | IQ3 | 0.891 | 2.690 | | | |
| | IQ4 | 0.756 | 1.494 | | | |
| TPACK | PCK | 0.863 | 2.275 | 0.875 | 0.915 | 0.728 |
| | TCK | 0.831 | 1.948 | | | |
| | TK | 0.868 | 2.282 | | | |
| | TPACK | 0.850 | 2.204 | | | |
| SEQ | SEQ1 | 0.929 | 3.970 | 0.932 | 0.951 | 0.830 |
| | SEQ2 | 0.906 | 3.338 | | | |
| | SEQ3 | 0.910 | 3.501 | | | |
| | SEQ4 | 0.900 | 3.141 | | | |
| SQ | SQ1 | 0.909 | 3.085 | 0.923 | 0.945 | 0.812 |
| | SQ2 | 0.894 | 3.028 | | | |
| | SQ3 | 0.898 | 3.158 | | | |
| | SQ4 | 0.904 | 3.256 | | | |
| US | US1 | 0.909 | 3.360 | 0.942 | 0.958 | 0.852 |
| | US2 | 0.929 | 4.243 | | | |
| | US3 | 0.925 | 4.088 | | | |
| | US4 | 0.928 | 4.202 | | | |
| USE | USE1 | 0.894 | 3.200 | 0.919 | 0.939 | 0.757 |
| | USE2 | 0.855 | 2.461 | | | |
| | USE3 | 0.853 | 2.512 | | | |
| | USE4 | 0.839 | 2.395 | | | |
| | USE5 | 0.906 | 3.575 | | | |

57–60]. It suggests that while service quality does not directly influence continuous use, it still indirectly impacts perceived satisfaction. Students' inclination to persist in using the online learning system is influenced by their satisfaction levels, which are in turn influenced by service quality.

Regarding hypothesis H3a and H3b, this study does not reveal a significant relationship between information quality and learners' perceived satisfaction or continuous use. These results differ from the findings of previous researchers such as [13, 21], who suggested a

**Table 3. Discriminant validity Fornell-Larcker criterion.**

| | IQ | NB | SEQ | SQ | TPACK | US | USE |
|---|---|---|---|---|---|---|---|
| IQ | **0.852** | | | | | | |
| NB | 0.088 | **0.894** | | | | | |
| SEQ | 0.411 | 0.160 | **0.911** | | | | |
| SQ | 0.467 | 0.231 | 0.685 | **0.901** | | | |
| TPACK | 0.398 | 0.320 | 0.446 | 0.528 | **0.853** | | |
| US | 0.205 | 0.601 | 0.350 | 0.390 | 0.327 | **0.923** | |
| USE | 0.121 | 0.586 | 0.263 | 0.344 | 0.336 | 0.603 | **0.870** |

**Table 4. HTMT.**

|  | IQ | NB | SEQ | SQ | TPACK | US | USE |
|---|---|---|---|---|---|---|---|
| IQ |  |  |  |  |  |  |  |
| NB | 0.099 |  |  |  |  |  |  |
| SEQ | 0.452 | 0.177 |  |  |  |  |  |
| SQ | 0.517 | 0.255 | 0.737 |  |  |  |  |
| TPACK | 0.455 | 0.365 | 0.493 | 0.587 |  |  |  |
| US | 0.225 | 0.662 | 0.372 | 0.416 | 0.359 |  |  |
| USE | 0.134 | 0.653 | 0.283 | 0.370 | 0.373 | 0.647 |  |

positive impact of information quality on perceived satisfaction and sustained use. This disparity may be attributed to the characteristics of open and distance education students, who often rely heavily on teachers and may lack active learning behaviors. Furthermore, learners' perceived satisfaction is influenced by their active cognition of information and the alignment of their needs and expectations with the available resources. The lack of significant association between information quality and perceived satisfaction in this study supports the notion that perceived satisfaction formation involves multiple comparisons between learners and information resources.

**Table 5. Cross loading.**

|  | IQ | NB | SEQ | SQ | TPACK | US | USE |
|---|---|---|---|---|---|---|---|
| IQ1 | **0.879** | 0.083 | 0.349 | 0.411 | 0.375 | 0.186 | 0.093 |
| IQ2 | **0.875** | 0.053 | 0.334 | 0.397 | 0.364 | 0.146 | 0.084 |
| IQ3 | **0.891** | 0.081 | 0.408 | 0.449 | 0.385 | 0.190 | 0.115 |
| IQ4 | **0.756** | 0.077 | 0.299 | 0.327 | 0.230 | 0.170 | 0.116 |
| AD | 0.077 | **0.880** | 0.135 | 0.203 | 0.290 | 0.537 | 0.522 |
| CD | 0.088 | **0.910** | 0.148 | 0.220 | 0.299 | 0.550 | 0.514 |
| PD | 0.071 | **0.893** | 0.146 | 0.196 | 0.269 | 0.526 | 0.537 |
| SEQ1 | 0.390 | 0.166 | **0.929** | 0.648 | 0.430 | 0.339 | 0.264 |
| SEQ2 | 0.380 | 0.150 | **0.906** | 0.615 | 0.412 | 0.322 | 0.227 |
| SEQ3 | 0.344 | 0.133 | **0.910** | 0.605 | 0.391 | 0.299 | 0.224 |
| SEQ4 | 0.381 | 0.131 | **0.900** | 0.626 | 0.392 | 0.313 | 0.241 |
| SQ1 | 0.450 | 0.241 | 0.639 | **0.909** | 0.495 | 0.382 | 0.359 |
| SQ2 | 0.427 | 0.179 | 0.602 | **0.894** | 0.486 | 0.338 | 0.279 |
| SQ3 | 0.394 | 0.173 | 0.603 | **0.898** | 0.460 | 0.326 | 0.301 |
| SQ4 | 0.410 | 0.232 | 0.622 | **0.904** | 0.462 | 0.353 | 0.293 |
| PCK | 0.355 | 0.274 | 0.390 | 0.454 | **0.863** | 0.290 | 0.291 |
| TCK | 0.308 | 0.287 | 0.318 | 0.413 | **0.831** | 0.252 | 0.293 |
| TK | 0.357 | 0.281 | 0.417 | 0.467 | **0.868** | 0.299 | 0.298 |
| TPACK | 0.336 | 0.247 | 0.398 | 0.470 | **0.850** | 0.272 | 0.263 |
| US1 | 0.156 | 0.525 | 0.414 | 0.410 | 0.285 | **0.909** | 0.574 |
| US2 | 0.197 | 0.546 | 0.289 | 0.334 | 0.304 | **0.929** | 0.549 |
| US3 | 0.218 | 0.574 | 0.301 | 0.344 | 0.315 | **0.925** | 0.540 |
| US4 | 0.188 | 0.575 | 0.285 | 0.349 | 0.302 | **0.928** | 0.561 |
| USE1 | 0.107 | 0.530 | 0.251 | 0.339 | 0.328 | 0.533 | **0.894** |
| USE2 | 0.111 | 0.523 | 0.216 | 0.310 | 0.314 | 0.537 | **0.855** |
| USE3 | 0.140 | 0.486 | 0.249 | 0.286 | 0.282 | 0.489 | **0.853** |
| USE4 | 0.079 | 0.480 | 0.182 | 0.251 | 0.221 | 0.524 | **0.839** |
| USE5 | 0.093 | 0.526 | 0.246 | 0.306 | 0.310 | 0.536 | **0.906** |

**Table 6. PLS-SEM path coefficients.**

| Path | Standardized coefficient | Std | t values | p values | Result |
|---|---|---|---|---|---|
| H1a: SQ -> USE | 0.126 | 0.047 | 2.665 | 0.008 | support |
| H1b: SQ -> US | 0.223 | 0.051 | 4.377 | *** | support |
| H2a: SEQ -> USE | -0.04 | 0.041 | 0.984 | 0.325 | not support |
| H2b: SEQ -> US | 0.134 | 0.042 | 3.171 | 0.002 | support |
| H3a: IQ -> US | -0.016 | 0.042 | 0.37 | 0.712 | not support |
| H3b: IQ -> USE | -0.09 | 0.047 | 1.93 | 0.054 | not support |
| H4a: TPACK -> US | 0.155 | 0.043 | 3.613 | *** | support |
| H4b: TPACK -> USE | 0.147 | 0.038 | 3.836 | *** | support |
| H4c: TPACK -> NB | 0.086 | 0.038 | 2.264 | 0.024 | support |
| H5: US -> USE | 0.538 | 0.044 | 12.216 | *** | support |
| H6: US -> NB | 0.373 | 0.062 | 5.992 | *** | support |
| H7: USE -> NB | 0.332 | 0.057 | 5.817 | *** | support |
| SRMR composite model = 0.036 | | | | | |
| $R^2_{NB}$ = 0.446; $Q^2_{NB}$ = 0.353 | | | | | |
| $R^2_{US}$ = 0.181; $Q^2_{US}$ = 0.151 | | | | | |
| $R^2_{USE}$ = 0.396; $Q^2_{USE}$ = 0.295 | | | | | |

Note: $^*p<0.05$, $^{**}p<0.01$, $^{***}p<0.001$.

Similarly, hypothesis H3b is not supported, indicating that information quality does not have a positive impact on continuous use. This finding can be attributed to the fact that students primarily utilize online learning systems for mandatory tasks, such as accessing teacher lectures, submitting assignments, or taking exams, rather than actively engaging with the system for their own learning purposes. This finding is consistent with the recent study by [11].

Conversely, the findings substantiate hypotheses H4a, H4b, and H4c, signifying that teachers' Technological Pedagogical Content Knowledge (TPACK) directly influences students' satisfaction with the learning system, their inclination to persist in using the system, and their educational benefits. These findings align with qualitative research conducted by [61, 62], which emphasized the positive impact of applying TPACK knowledge in practical teaching and instructional design. This study contributes to the extant literature by furnishing quantitative evidence that substantiates the correlation between teachers' TPACK proficiency and students' satisfaction with learning, their intent to continue learning, and the benefits derived from learning. It is worth noting that limited quantitative research has explored this relationship before.

Furthermore, some empirical studies have demonstrated the positive impact of teachers' TPACK ability on continuous use intention [34] and students' willingness to continue learning and academic performance [39]. These studies further validate the significance of teachers' TPACK knowledge in enhancing students' learning outcomes and their engagement with online learning systems.

The findings also support hypothesis H5, indicating that user satisfaction (US) has a significant positive effect on continuous use of online learning. This result is consistent with the conclusions drawn by [11, 42, 63, 64]. This implies that students' perceived satisfaction significantly influences their continuous engagement with the online learning system.

Lastly, hypothesis H6 is supported, as this study aligns with previous research by highlighting that higher perceived satisfaction is associated with improved learning benefits for students [11, 13, 14, 22]. Conversely, lower satisfaction levels are linked to a decline in students'

learning benefits. These findings emphasize the importance of fostering students' satisfaction in order to maximize their learning outcomes.

Additionally, the study supports hypothesis H7, affirming the advantages of sustained use of online learning systems. When these systems effectively cater to students' needs, there's a higher likelihood of continued usage, fostering effective and successful learning experiences. These findings resonate with prior literature, encompassing studies by [11, 13, 14, 22, 65].

In summary, this study contributes to comprehending the factors that influence learners' satisfaction, continuous use, and learning benefits within the online learning context. The study's findings align with previous research, revealing both consistencies and discrepancies. These outcomes offer valuable insights for educational institutions and online learning providers to enhance online learning system design and delivery, improve service quality, and emphasize teachers' TPACK knowledge, ultimately optimizing students' learning experiences.

## Implications

Firstly, it introduces a comprehensive multidimensional model for adult online learning (OL) success in Western China, amalgamating ISS and TPACK theories. This model, built upon prior research literature, extends the scope of influencing factors, emphasizing system quality, service quality, and teachers' TPACK ability. Empirical validation within this model confirms these factors' association with perceived satisfaction, continuous use, and overall benefits in adult OL.

Secondly, prior research has primarily concentrated on understanding the correlation between factors affecting perceived satisfaction and continued use, overlooking the pivotal role of teachers' TPACK ability in OL success. This study unveils the significant positive impact of TPACK on learners' continuous system usage, perceived satisfaction, and OL benefits. Stronger TPACK correlates with increased OL system usage and heightened satisfaction levels, ultimately leading to enhanced learning benefits. This clarification of TPACK's direct influence on OL benefits stands as the second significant contribution.

Thirdly, while previous empirical studies predominantly centered on full-time students, this research pivots its focus to working adult learners who encounter unique pressures from work, family, and societal obligations. The study challenges prior conclusions by revealing an absence of a positive link between learner quality and perceived satisfaction among adult learners. This divergence likely stems from the different research subjects. Within the context of adult learners, this study incorporates cultural and grit factors affecting OL success. It establishes positive correlations between grit and perceived satisfaction, continuous use, and the moderating effects of long/short-term orientation on the relationship between continuous use and benefits. This focused approach represents the third valuable contribution, offering more targeted and pertinent insights.

Fourthly, this study is to shift the focus of OL success research from developed countries to developing countries, particularly in China's southwestern region. Although OL is applied globally, different factors such as OL environment, cultural background, education system, and economic conditions in different regions may affect students' OL success. Therefore, it is necessary to conduct research on OL success in developing countries. By studying the factors influencing adult OL success in China's southwestern region through distance open education, this study has filled the research gap in this field in developing countries. This will help provide more scientific and feasible strategies for OL success for China and other developing countries' online education. This study not only thoroughly explored the influencing factors of OL success at the theoretical level but also provided specific suggestions and solutions at the practical level, providing a reference for the improvement and development of distance open education teaching in this region.

## Conclusion

In summary, this study presents a comprehensive model for adult online learning success, amalgamating the ISS Model and TPACK. The findings underscore the pivotal role of a teacher's instructional prowess, particularly their TPACK, in augmenting the benefits derived from adult learners' online education experiences. The study accentuates the paramount importance of teachers' TPACK competency in shaping online learning success. Results signify that a robust TPACK proficiency positively impacts learners' sustained engagement and perceived satisfaction. Moreover, the study underscores the positive instructional influence of TPACK on students' learning outcomes within an online setting. Strengthening TPACK skills holds promise in improving pedagogical practices, fostering continued use of online learning platforms, and elevating learner satisfaction, particularly among adult learners in this study.

However, it's essential to acknowledge certain limitations. Firstly, this study only treated students as respondents and did not consider the perspectives of teachers and administrators, who also Significantly contribute to the OL process. Therefore, subsequent studies should investigate the perspectives of instructors and administrators to examine the factors that contribute to the achievements of adult online learners. Secondly, the sample of this study was limited to learners in certain regions, so caution should be exercised when generalizing the results. As different regions have different learning environments and resources, more research is needed to determine the success factors of adult OL in different areas.

## Future research

Future research should broaden its scope to enhance the validation and reliability of this model. This model, uniting the TPACK theory with the ISS model, outlines adult online learning success in Western China. Yet, there is a necessity to delve deeper into the diverse perspectives influencing success factors in adult online learning.

## Supporting information

**S1 Data.**
(CSV)

**S1 Appendix. The items with their sources.**
(DOCX)

## Acknowledgments

The authors would like to acknowledge the Universiti Malaya for the financial support through the University Grant -UMG0030-2021 (UM.0000412/HGA.GV).

## Author Contributions

**Supervision:** Rafiza Abdul Razak.

**Writing – original draft:** Li Yuebo.

**Writing – review & editing:** Siti Hajar Halili.

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
