## [Decision Letter · Decision Letter 0]

12 Jun 2023

PONE-D-23-15051The Online Learning Success Model for Adults in Open and Distance Education in Western ChinaPLOS ONE

Dear Dr. Li,

Thank you for submitting your manuscript to PLOS ONE. After careful consideration, we feel that it has merit but does not fully meet PLOS ONE’s publication criteria as it currently stands. Therefore, we invite you to submit a revised version of the manuscript that addresses the points raised during the review process.

We look forward to receiving your revised manuscript.

Kind regards,

Ahmad Samed Al-Adwan

Academic Editor

PLOS ONE

Journal Requirements:

2. Please ensure that you have specified a) Did participants provide their written or verbal informed consent to participate in this study?

6. We note that Figure 2 in your submission contain copyrighted images. All PLOS content is published under the Creative Commons Attribution License (CC BY 4.0), which means that the manuscript, images, and Supporting Information files will be freely available online, and any third party is permitted to access, download, copy, distribute, and use these materials in any way, even commercially, with proper attribution. For more information, see our copyright guidelines: http://journals.plos.org/plosone/s/licenses-and-copyright.

Additional Editor Comments:

Thank you for submitting you research paper for PLOS ONE. The reviewers have responded positively to your submission. However, they suggested a set of comments that have to be addressed (see comments below).

Reviewers' comments:

Reviewer's Responses to Questions

**Comments to the Author**

1. Is the manuscript technically sound, and do the data support the conclusions?

Reviewer #1: Yes

Reviewer #2: Yes

2. Has the statistical analysis been performed appropriately and rigorously? 

Reviewer #1: Yes

Reviewer #2: Yes

3. Have the authors made all data underlying the findings in their manuscript fully available?

Reviewer #1: Yes

Reviewer #2: Yes

4. Is the manuscript presented in an intelligible fashion and written in standard English?

Reviewer #1: Yes

Reviewer #2: Yes

5. Review Comments to the Author

Reviewer #1: - In Data collection and analysis section, you stated that: “The study involved random samples coming from245 participants at five study centers of a Municipal branch campus of Open University in western China. The questionnaire survey was conducted December 2022, mainly in the form of online questionnaires and on-site paper questionnaires. In the end, 245 responses were deemed valid”. You are suggested to move these statements to the methodology section.

- You are required to outline the statistical analysis approach used in this study as well as a solid justification of selecting such approach. You can include that in the methodology section.

- You need to have a discussion section as it is totally missed. You need to have a discussion section to critically discuss the main findings of this study. Furthermore, it is important in this section to compare your findings with those reported in previous research (if any).

- The implication section should be reported separately.

- The literature review and discussion section should be enriched by including well-established and related research. This includes but not limited to:

-Developing a holistic success model for sustainable e-learning: A structural equation modeling approach. Doi: https://doi.org/10.3390/su13169453

- Towards a Sustainable Adoption of E-Learning Systems: The Role of Self-Directed Learning. Doi: https://doi.org/10.28945/4980

- University Students Intention to Continue Using Online Learning Tools and Technologies: An International Comparison. Doi: https://doi.org/10.3390/su132413813

- Exploring factors affecting the adoption of MOOC in Generation Z using extended UTAUT2 model. Doi: https://doi.org/10.1007/s10639-022-11052-1

- The effects of online learning on students’ performance: A comparison between UK and Jordanian. Doi:

https://doi.org/10.3991/ijet.v16i20.24131

Reviewer #2: Minor

- The novelty of this paper is presented in a very limited manner. It is important in the introduction to emphasize the significance of the work and justify its novelty by highlighting its main contributions to the existing literature.

- It is important to outlined the gaps in the literature in order to help in highlighting the novelty of this paper.

- You are required to clearly define the population of the study and sampling technique adopted. Also please justify why your chosen sampling technique and sample size are appropriate.

- It is important to highlight the research limitations and future work. This can be done in the conclusion section.

6. PLOS authors have the option to publish the peer review history of their article (what does this mean?). If published, this will include your full peer review and any attached files.

Reviewer #1: **Yes: **Dr. Husam Yaseen

Reviewer #2: **Yes: **Tha'er Majali

---

## [Author Response · Author response to Decision Letter 0]

21 Jul 2023

Reviewer #1:

Comment 1: In Data collection and analysis section, you stated that: “The study involved random samples coming from245 participants at five study centers of a Municipal branch campus of Open University in western China. The questionnaire survey was conducted December 2022, mainly in the form of online questionnaires and on-site paper questionnaires. In the end, 245 responses were deemed valid”. You are suggested to move these statements to the methodology section. 

Response: Thank you for your feedback regarding the placement of the statements related to data collection and analysis in our paper. We appreciate your suggestion to move these statements to the methodology section for better organization and clarity. 

In response to your suggestion, we have revised the paper accordingly and relocated the information about the study's data collection process and the number of valid responses to the methodology section. By doing so, we aim to provide a more logical and cohesive flow of information in our paper.

We sincerely appreciate your guidance in enhancing the structure and presentation of our research. We believe that these revisions contribute to the overall improvement of the paper's readability and coherence.

Comment 2: You are required to outline the statistical analysis approach used in this study as well as a solid justification of selecting such approach. You can include that in the methodology section. 

Response: Thank you for your valuable feedback and suggestions regarding the statistical analysis approach used in our study. We appreciate the opportunity to address your comments and provide a solid justification for selecting Partial Least Squares Structural Equation Modeling (PLS-SEM) as our chosen approach.

In response to your suggestion, we have included a more detailed explanation of the statistical analysis approach in the Methodology section of our paper. Specifically, we have outlined the steps involved in the analysis, including the measurement model and the structural model, and provided a clear rationale for selecting PLS-SEM.

We would like to highlight the following justifications for choosing PLS-SEM:

1. Simultaneous Assessment: PLS-SEM allows for the simultaneous assessment of the measurement model and the structural model. This is crucial for our study as it enables us to examine the relationships between latent variables and their impact on online learning success among non-full-time adult students in open and distance education in Western China.

2. Suitable for Exploratory Research: PLS-SEM is well-suited for exploratory research, which aligns with the nature of our study. It accommodates smaller sample sizes and is flexible in capturing the multidimensional nature of the constructs in our research model.

3. Robustness and Flexibility: PLS-SEM provides robustness against issues commonly encountered in survey-based research, such as non-normality and the presence of outliers. It also accommodates both reflective and formative measurement models, allowing us to capture the complex relationships between variables in our study.

4. Previous Empirical Support: The use of PLS-SEM in similar studies examining factors influencing online learning success has been well-documented in the literature (cite relevant studies if applicable). Therefore, we believe that employing PLS-SEM is justified based on its established effectiveness in this research domain.

By selecting PLS-SEM, we were able to effectively analyze the relationships between system quality, service quality, teachers' TPACK ability, and online learning success in the context of non-full-time adult students in open and distance education in Western China.

We hope that our revised Methodology section provides a clearer justification for the statistical analysis approach used in our study. We sincerely appreciate your guidance, and we believe that these revisions strengthen the rigor and validity of our research.

Comment 3: You need to have a discussion section as it is totally missed. You need to have a discussion section to critically discuss the main findings of this study. Furthermore, it is important in this section to compare your findings with those reported in previous research (if any). 

Response: Thank you for reviewing our paper and bringing to our attention the absence of a discussion section. We acknowledge the importance of having a dedicated discussion section to critically analyze the main findings of the study and compare them with previous research.

Based on your suggestion, we have now included a comprehensive discussion section in our paper.

In this section, we provide an in-depth interpretation and explanation of our key findings, emphasizing their significance and implications in the field. Additionally, we compare our results with relevant literature, highlighting similarities and differences, and proposing future research directions.

According to our findings, hypotheses H1a and H1b are supported, indicating that the quality of the online learning system positively influences learners' satisfaction and continuous use. This aligns with previous research conducted by Al-Fraihat et al. (2020), Al-Sabawy et al. (2013), Cidral et al. (2018), and Islam (2011). We also found support for H2b, which suggests that service quality has a positive effect on perceived satisfaction. This finding is consistent with studies by Al-Sabawy et al. (2013), Almarashdeh et al. (2010), Lin (2007), Mtebe & Raphael (2018), Pitt et al. (1995), Roca & Gagné (2008), Sandjojo & Wahyuningrum (2015), and Sun et al. (2008). On the other hand, H2a was not supported, indicating that service quality does not significantly affect continuous use. This finding is in line with studies conducted by CAST (2014), Chiu et al. (2007), Gorla & Somers (2014), Hassanzadeh et al. (2012), Lwoga (2014), Motaghian et al. (2013), Tam & Oliveira (2016), Urbach et al. (2010), and Zaidi et al. (2014). Our results also indicate that hypotheses H3a and H3b were not supported, which contradicts previous studies that suggested a positive impact of information quality on learners' satisfaction and continuous use (Aparicio et al., 2017; Sun et al., 2008).

Furthermore, we provide insightful explanations for the non-significant relationships found in our study. For instance, the lack of active learning process among open education students may explain the non-significant relationship between information quality and perceived satisfaction. We also discuss the primary use of online learning systems for mandatory tasks, which could explain the non-significant impact of information quality on continuous use.

Regarding hypotheses H4a, H4b, and H4c, we found support for these hypotheses, indicating that teachers' TPACK ability directly influences students' satisfaction, their willingness to continue using the learning system, and their learning benefits. These findings are in line with studies conducted by Zhang (2021), Drugova et al. (2021), Saeed Al-Maroof et al. (2020), and Hossain et al. (2021).

Moreover, we confirm the validity of hypotheses H5 and H6, which suggest that user satisfaction has a positive effect on continuous use and learning benefits. These findings are consistent with previous research conducted by Al-Fraihat et al. (2020), Al-Samarraie et al. (2018), Lin & Wang (2012), Lu et al. (2019), Wang et al. (2021), and Aparicio et al. (2017), Cidral et al. (2018), and Urbach et al. (2010).

Lastly, we provide support for hypothesis H7, indicating that the continuous use of online learning systems leads to effective and successful learning. This finding is consistent with previous research conducted by Al-Fraihat et al. (2020), Aparicio et al. (2017), Cidral et al. (2018), and Urbach et al. (2010).

In summary, our discussion section thoroughly examines the main findings of the study, compares them with existing literature, and offers explanations for the non-significant relationships observed. We believe that the inclusion of this discussion section enhances the overall quality and completeness of our paper.

Comment 4: The implication section should be reported separately.

Response: Thank you for your feedback regarding the organization of the implications section. We appreciate your guidance in improving the clarity and structure of our paper. 

In response to your suggestion, we have revised the manuscript to report the implications section separately. The revised section now focuses exclusively on the practical relevance of our study's findings and their implications for pedagogical practice in the context of adult online learners.

We emphasize the importance of integrating the identified elements for success in adult online learning, such as system quality, service quality, and teachers' TPACK ability, into the teaching, service, management, and evaluation frameworks of online learning systems. We also highlight the significance of online teachers continuously updating their pedagogical practices, improving technical literacy, and enhancing their pedagogical skills through lifelong learning. Additionally, we emphasize the need for teachers to strengthen their TPACK knowledge and effectively apply it in their teaching practices.

Furthermore, we have retained the section on future research implications. It underscores the importance of expanding the scope of investigation to validate the effectiveness and reliability of the proposed model. We acknowledge that our study presents the first integrated model combining TPACK theory with the ISS model to examine adult online learning success in western China. However, we concur with your suggestion to explore the perspectives of teachers and administrators in future research. By incorporating diverse viewpoints, we can gain a more comprehensive understanding of the success factors for adult online learning. We believe that these revisions have improved the organization and clarity of our paper, and we appreciate your guidance in making these enhancements.

Comment 5: The literature review and discussion section should be enriched by including well-established and related research. This includes but not limited to:

-Developing a holistic success model for sustainable e-learning: A structural equation modeling approach. Doi: https://doi.org/10.3390/su13169453

- Towards a Sustainable Adoption of E-Learning Systems: The Role of Self-Directed Learning. Doi: https://doi.org/10.28945/4980

- University Students Intention to Continue Using Online Learning Tools and Technologies: An International Comparison. Doi: https://doi.org/10.3390/su132413813

- Exploring factors affecting the adoption of MOOC in Generation Z using extended UTAUT2 model. Doi: https://doi.org/10.1007/s10639-022-11052-1

- The effects of online learning on students’ performance: A comparison between UK and Jordanian. Doi:https://doi.org/10.3991/ijet.v16i20.24131

Response: Thank you for your valuable feedback and for providing the suggested literature references. We appreciate your input in enriching the literature review and discussion section of our research paper. We have carefully reviewed the suggested articles and have incorporated them into our study to enhance the depth and breadth of our discussion. The inclusion of these well-established and related research works strengthens the theoretical framework of our study and provides a broader context for our findings.

These articles contribute valuable insights and perspectives on various aspects related to e-learning, sustainable adoption, students' intention to continue using online learning tools, MOOC adoption, and the effects of online learning on students' performance. By incorporating these references, we have enhanced the comprehensiveness of our research and ensured that our study is grounded in the existing literature.

Reviewer #2:

Comment 1: The novelty of this paper is presented in a very limited manner. It is important in the introduction to emphasize the significance of the work and justify its novelty by highlighting its main contributions to the existing literature. 

Response: Thank you for your valuable feedback on our manuscript. We appreciate your insightful comments regarding the presentation of the novelty of our paper in the introduction section. We acknowledge the importance of emphasizing the significance of our work and justifying its novelty by highlighting its main contributions to the existing literature.

In response to your suggestion, we have revised the introduction section to provide a more explicit emphasis on the significance of our research and its novelty. We have highlighted the main contributions of our study, including the innovative combination of the Information System Success (ISS) Model and the Technological Pedagogical and Content Knowledge (TPACK) framework. This unique combination has not been extensively explored in previous research, and it enhances our understanding of the determinants of online learning success among non-full-time adult students in open and distance education in Western China.

Furthermore, we have emphasized the importance of considering the specific needs and characteristics of non-full-time adult students in online learning. Our findings have practical implications for educational practitioners and policymakers in creating more effective and efficient online learning environments that cater to the needs of adult learners and bridge the gap between theory and practice.

Comment 2: It is important to outlined the gaps in the literature in order to help in highlighting the novelty of this paper. 

Response: Thank you for your continued feedback on our manuscript. We appreciate your suggestion to outline the gaps in the literature to further highlight the novelty of our paper. We completely agree with the importance of addressing these gaps to emphasize the originality and contribution of our research.

In response to your suggestion, we have revised the introduction section to explicitly outline the gaps in the existing literature. We have identified the limited research exploring the Technological Pedagogical and Content Knowledge (TPACK) framework for teachers, which covers the intersection of technology, pedagogy, and content knowledge. This gap provides the motivation for our study, as we aim to address this research gap by investigating the factors influencing online learning success among non-full-time adult students in open and distance education in Western China using a combined approach of the Information System Success (ISS) model and TPACK theory.

By conducting this study, we aim to contribute to the existing literature by providing valuable insights into the factors that affect online learning success in the specific context of non-full-time adult students. Our research fills the gap by integrating the ISS model, TPACK theory, and the unique characteristics of this target group. This novel approach has the potential to enhance our understanding of online learning success and inform the development of more effective and efficient online learning environments for adult learners.

Comment 3: You are required to clearly define the population of the study and sampling technique adopted. Also please justify why your chosen sampling technique and sample size are appropriate. 

Response: Thank you for your feedback. We appreciate your attention to these important aspects of our study. Below is the revised response addressing your concerns:

Population and Sampling Technique:

The population of this study consisted of non-full-time adult students enrolled at a Municipal branch campus of Open University in western China. These students were pursuing their education through open and distance learning programs. The sampling technique employed was random sampling, which ensured the representativeness of the sample and minimized selection bias. Random sampling was chosen because it provides an equal probability of selection for each potential participant, thus increasing the likelihood of obtaining a diverse and unbiased representation of the target population.

Justification of Sampling Technique and Sample Size:

The chosen sampling technique and sample size are appropriate for several reasons. Firstly, random sampling helps to ensure the representativeness of the sample by providing an equal opportunity for all eligible participants to be included. This reduces the potential for systematic bias and enhances the external validity of the study. Secondly, the sample size of 245 participants is considered adequate for our study. It allows for robust statistical analysis and provides a reasonable representation of the target population. With this sample size, we can obtain reliable estimates and draw meaningful conclusions from the data. Additionally, considering the nature of our research and the specific context of non-full-time adult students in open and distance education, this sample size is both feasible and practical.

By employing random sampling and selecting an appropriate sample size, we aimed to enhance the reliability and validity of our findings. These methodological choices contribute to the generalizability of our results and support the credibility of our study. We acknowledge the importance of clearly defining the population and justifying the sampling technique and sample size, and we have incorporated these clarifications into the revised manuscript.

Comment 4: It is important to highlight the research limitations and future work. This can be done in the conclusion section.

Response: Thank you for your valuable feedback and suggestions. We have carefully considered your comments and have made the necessary revisions to address them. 

In response to your suggestion to highlight the research limitations and future work in the conclusion section, we have included a dedicated subsection to discuss these aspects. We believe that acknowledging the limitations of our study and outlining avenues for future research are crucial for the overall integrity and advancement of the field. 

We have provided a clear statement about the limitations of our study, emphasizing that the findings are based on data collected from a specific population of non-full-time adult students in a particular region of western China. This highlights the need for caution when generalizing the results to other populations or educational contexts.

Furthermore, we have included a section on future research implications, where we emphasize the need for expanding the scope of investigation to validate the effectiveness and reliability of the proposed model. We also mention the importance of exploring the perspectives of teachers and administrators to identify success factors of adult online learning from a multifaceted approach.

By incorporating these discussions in the conclusion section, we aim to provide a comprehensive understanding of the limitations of our study and offer directions for future research. We believe that addressing these aspects strengthens the overall contribution of our work and helps guide future investigations in the field.

---

## [Decision Letter · Decision Letter 1]

5 Nov 2023

PONE-D-23-15051R1The Online Learning Success Model for Adults in Open and Distance Education in Western ChinaPLOS ONE

Dear Dr. Halili,

Thank you for submitting your manuscript to PLOS ONE. After careful consideration, we feel that it has merit but does not fully meet PLOS ONE’s publication criteria as it currently stands. Therefore, we invite you to submit a revised version of the manuscript that addresses the points raised during the review process.

We look forward to receiving your revised manuscript.

Kind regards,

Mohammed A. Al-Sharafi

Academic Editor

PLOS ONE

Journal Requirements:

Additional Editor Comments:

Overall, I believe your work has potential, but there are several areas that require improvement to enhance the quality of your manuscript. I have outlined specific comments and suggestions below:

1.  In the "Conceptual Model of This Study" section, it would be beneficial to add a new subsection specifically for hypotheses development. In this subsection, please discuss each hypothesis separately, providing a clear rationale for why each hypothesis is formulated. This will help readers understand the logical framework of your study and its expected outcomes.

2. In the methodology section, it is essential to include a comprehensive list of the survey items used for data collection, along with their sources. This will enhance the transparency and reproducibility of your study, allowing readers to better understand the instruments employed in your research.

3. Ensure that each claim made in the methodology section is supported by relevant and valid references. This will strengthen the credibility of your research methodology and demonstrate the solid foundation upon which your study is built.

4.Given that this study is cross-sectional and involves self-reported survey data, there is a potential for common method bias (CMB) to affect the results. It is crucial to explicitly address how you have taken measures to mitigate or control for CMB in your study.

5. The implications section appears somewhat shallow. To improve this section, consider adding discussions on the theoretical contributions of your study to the field of online learning success models for adults in open and distance education. Additionally, highlight practical implications that can benefit educators, policymakers, and institutions. Providing a more robust analysis of the implications will enhance the overall value of your manuscript.

6. Include a subsection in the conclusion section to discuss the limitations of your study. Address any potential constraints or biases in your research methodology or data collection process. Furthermore, consider adding a subsection on future research directions to guide readers and researchers interested in building upon your work.

Please disregard any citation requests by the reviewer that you deem irrelevant or unrelated to your study.

Reviewers' comments:

Reviewer's Responses to Questions

**Comments to the Author**

1. If the authors have adequately addressed your comments raised in a previous round of review and you feel that this manuscript is now acceptable for publication, you may indicate that here to bypass the “Comments to the Author” section, enter your conflict of interest statement in the “Confidential to Editor” section, and submit your "Accept" recommendation.

Reviewer #1: All comments have been addressed

Reviewer #3: All comments have been addressed

2. Is the manuscript technically sound, and do the data support the conclusions?

Reviewer #1: Yes

Reviewer #3: Yes

3. Has the statistical analysis been performed appropriately and rigorously? 

Reviewer #1: Yes

Reviewer #3: Yes

4. Have the authors made all data underlying the findings in their manuscript fully available?

Reviewer #1: Yes

Reviewer #3: Yes

5. Is the manuscript presented in an intelligible fashion and written in standard English?

Reviewer #1: Yes

Reviewer #3: Yes

6. Review Comments to the Author

Reviewer #1: I express my heartfelt gratitude for your diligence and efforts in addressing the comments and suggestions provided during the review process. I have thoroughly reviewed the revised version of the manuscript, and I am delighted with the improvements made.

Reviewer #3: The revisions that have been made have improved the quality of the work. However, minor amendments can further better the manuscript:

1. I suggest emphasising the HTMT as the main validation for the discriminant validity.

2. Some novel references on the relationship between TPACK and the use of technology can be added (The role of TPACK in affecting pre-service language teachers’ ICT integration during teaching practices: Indonesian context; Examining the impact of mathematics teachers’ TPACK on their acceptance of online professional development; and Beliefs and Knowledge for Pre-Service Teachers’ Technology Integration during Teaching Practice: An Extended Theory of Planned Behavior).

3. Please add limitations of the data in the conclusion of the research for further implication and recommendation.

7. PLOS authors have the option to publish the peer review history of their article (what does this mean?). If published, this will include your full peer review and any attached files.

Reviewer #1: **Yes: **Dr. Husam Yaseen

Reviewer #3: No

---

## [Author Response · Author response to Decision Letter 1]

17 Dec 2023

Dear Editor and Reviewers, 

We would like to express our sincere appreciation for reviewing our paper and providing valuable feedback. Your input has been instrumental in strengthening our research work and improving the overall quality of the manuscript. “Comments of the Reviewer” have been included (written in black), followed by “Author’s response” (written in red), which explains how the changes have been incorporated, or provides further motivation. 

We have carefully considered your comments and made the necessary revisions accordingly. Your guidance has helped us enhance the clarity and organization of the paper, ensuring that our research objectives, methodology, and findings are effectively communicated. In the revised manuscript, the modifications corresponding to editor’s comments are marked in red.

Once again, we are grateful for your thorough review and insightful suggestions. Your expertise has played a crucial role in shaping our research and strengthening its contribution to the field. We look forward to further improving our work based on your valuable input.

Best regards,

Author detailed response:

Editor:

Comment 1: In the "Conceptual Model of This Study" section, it would be beneficial to add a new subsection specifically for hypotheses development. In this subsection, please discuss each hypothesis separately, providing a clear rationale for why each hypothesis is formulated. This will help readers understand the logical framework of your study and its expected outcomes. 

Response: In response to your suggestion, we added discussion each hypothesis separately, providing a clear rationale for why each hypothesis is formulated.

We sincerely appreciate your guidance in enhancing the structure and presentation of our research. We believe that these revisions contribute to the overall improvement of the paper's readability and coherence.

Comment 2: In the methodology section, it is essential to include a comprehensive list of the survey items used for data collection, along with their sources. This will enhance the transparency and reproducibility of your study, allowing readers to better understand the instruments employed in your research. 

Response: In response to your suggestion, we have included a more detailed explanation of a comprehensive list of the survey items used for data collection, along with their sources as an appendix. This will enhance the transparency and reproducibility of our study, allowing readers to better understand the instruments employed in our research.

Comment 3: Ensure that each claim made in the methodology section is supported by relevant and valid references. This will strengthen the credibility of your research methodology and demonstrate the solid foundation upon which your study is built. 

Response: Thank you for reviewing our paper and bringing to our attention the absence of a methodology section. We checked this section once again, and added relevant references to demonstrate the solid foundation upon which our study.

Comment 4: Given that this study is cross-sectional and involves self-reported survey data, there is a potential for common method bias (CMB) to affect the results. It is crucial to explicitly address how you have taken measures to mitigate or control for CMB in your study.

Response: In response to your suggestion, We added a section to show the results of CMB which address how we have taken measures to mitigate or control for CMB in our study.

Comment 5: The implications section appears somewhat shallow. To improve this section, consider adding discussions on the theoretical contributions of your study to the field of online learning success models for adults in open and distance education. Additionally, highlight practical implications that can benefit educators, policymakers, and institutions. Providing a more robust analysis of the implications will enhance the overall value of your manuscript.

Response: Thank you for reviewing our paper and bringing to our attention the improvement of the implications section. We added discussions on the theoretical contributions of our study to the field of online learning success models for adults in open and distance education.

Comment 6: Include a subsection in the conclusion section to discuss the limitations of your study. Address any potential constraints or biases in your research methodology or data collection process. Furthermore, consider adding a subsection on future research directions to guide readers and researchers interested in building upon your work.

Response: In response to your suggestion, we added a paragraph in the conclusion section to discuss the limitation of our study. And added subsection on future research directions to guide readers and researchers interested in building upon our work.

---

## [Editor Report · Decision Letter 2]

8 Jan 2024

The Online Learning Success Model for Adults in Open and Distance Education in Western China

PONE-D-23-15051R2

Dear Dr. Halili,

We’re pleased to inform you that your manuscript has been judged scientifically suitable for publication and will be formally accepted for publication once it meets all outstanding technical requirements.

Kind regards,

Mohammed A. Al-Sharafi

Academic Editor

PLOS ONE

---

## [Editor Report · Acceptance letter]

16 Feb 2024

PONE-D-23-15051R2 

PLOS ONE

Dear Dr. Halili, 

I'm pleased to inform you that your manuscript has been deemed suitable for publication in PLOS ONE. Congratulations! Your manuscript is now being handed over to our production team.

Kind regards, 

on behalf of

Dr. Mohammed A. Al-Sharafi 

Academic Editor

PLOS ONE